# Evaluation of antigen-detecting and antibody-detecting diagnostic test combinations for diagnosing melioidosis

**Premjit Amornchai**[1]*, **Viriya Hantrakun**[1], **Gumphol Wongsuvan**[1],
**Vanaporn Wuthiekanun**[1], **Surasakdi Wongratanacheewin**[2], **Prapit Teparrakkul**[3],
**T. Eoin West**[4,5], **David P. AuCoin**[6], **Nicholas P. J. Day**[1,7], **Paul J. Brett**[6], **Mary N. Burtnick**[6], **Narisara Chantratita**[1,5], **Direk Limmathurotsakul**[1,7,8]

**1** Mahidol Oxford Tropical Medicine Research Unit, Faculty of Tropical Medicine, Mahidol University, Bangkok, Thailand, **2** Department of Microbiology, Faculty of Medicine and Melioidosis Research Center, Khon Kaen University, Khon Kaen, Thailand, **3** Medical Department, Sunpasitthiprasong Hospital, Ubon Ratchathani, Thailand, **4** Division of Pulmonary and Critical Care Medicine, Harborview Medical Center, University of Washington, Seattle, Washington, United States of America, **5** Department of Microbiology and Immunology, Faculty of Tropical Medicine, Mahidol University, Bangkok, Thailand, **6** Department of Microbiology and Immunology, University of Nevada, Reno School of Medicine, Reno, Nevada, United States of America, **7** Centre for Tropical Medicine and Global Health, University of Oxford, Oxford, United Kingdom, **8** Department of Tropical Hygiene, Faculty of Tropical Medicine, Mahidol University, Bangkok, Thailand

* kung@tropmedres.ac

**Data Availability Statement:** The final databases with the data dictionary are publicly available online (https://doi.org/10.6084/m9.figshare.14345315).

## Abstract

### Background

Melioidosis, an infectious disease caused by *Burkholderia pseudomallei*, is endemic in many tropical developing countries and has a high mortality. Here we evaluated combinations of a lateral flow immunoassay (LFI) detecting *B. pseudomallei* capsular polysaccharide (CPS) and enzyme-linked immunosorbent assays (ELISA) detecting antibodies against hemolysin co-regulated protein (Hcp1) or O-polysaccharide (OPS) for diagnosing melioidosis.

### Methodology/Principal findings

We conducted a cohort-based case-control study. Both cases and controls were derived from a prospective observational study of patients presenting with community-acquired infections and sepsis in northeast Thailand (Ubon-sepsis). Cases included 192 patients with a clinical specimen culture positive for *B. pseudomallei*. Controls included 502 patients who were blood culture positive for *Staphylococcus aureus*, *Escherichia coli* or *Klebsiella pneumoniae* or were polymerase chain reaction assay positive for malaria or dengue. Serum samples collected within 24 hours of admission were stored and tested using a CPS-LFI, Hcp1-ELISA and OPS-ELISA. When assessing diagnostic tests in combination, results were considered positive if either test was positive. We selected ELISA cut-offs corresponding to a specificity of 95%. Using a positive cut-off OD of 2.912 for Hcp1-ELISA, the combination of the CPS-LFI and Hcp1-ELISA had a sensitivity of 67.7% (130/192 case patients) and a specificity of 95.0% (477/502 control patients). The sensitivity of the combination

**Funding:** The study was funded by the Wellcome Trust (090219/Z/09/Z and 220211/A/20/Z) to DL, National Heart, Lung and Blood Institute, National Institutes of Health (R01HL113382) to TEW, and National Science and Technology Development Agency (NSTDA) (P-16-51225) to SW. DL was supported by an intermediate fellowship from the Wellcome Trust (101103/Z/13/Z). The AMD LFI tests were kindly provided by InBios International with funding (1R42AI102482 to DA) from the National Institute of Allergy and Infectious Diseases. For the purpose of open access, the author has applied a CC BY public copyright licence to any Author Accepted Manuscript version arising from this submission. The funders had no role in the study design, data collection and analysis, decision to publish or preparation of the manuscript.

**Competing interests:** The authors have declared that no competing interests exist.

(67.7%) was higher than that of the CPS-LFI alone (31.3%, p<0.001) and that of Hcp1-ELISA alone (53.6%, p<0.001). A similar phenomenon was also observed for the combination of CPS-LFI and OPS-ELISA. In case patients, positivity of the CPS-LFI was associated with a short duration of symptoms, high modified Sequential (sepsis-related) Organ Failure Assessment (SOFA) score, bacteraemia and mortality outcome, while positivity of Hcp1-ELISA was associated with a longer duration of symptoms, low modified SOFA score, non-bacteraemia and survival outcome.

## Conclusions/Significance

A combination of antigen-antibody diagnostic tests increased the sensitivity of melioidosis diagnosis over individual tests while preserving high specificity. Point-of-care tests for melioidosis based on the use of combination assays should be further developed and evaluated.

## Author summary

Melioidosis is an infection caused by the Gram-negative bacterium *Burkholderia pseudomallei*. There are currently no commercially available and reliable point-of-care diagnostic tests for melioidosis. We previously demonstrated that a prototype lateral flow immunoassay (LFI) developed to detect *B. pseudomallei* capsular polysaccharide (CPS) had limited sensitivity (31.3%) but high specificity (98.8%) for diagnosing melioidosis among patients presenting with community-acquired infection or sepsis in northeast Thailand. Here, we evaluated combinations of the CPS-LFI and enzyme-linked immunosorbent assays (ELISA) that detect antibodies against hemolysin co-regulated protein (Hcp1) or O-polysaccharide (OPS). When used in combination, results were considered positive if either test was positive. We selected ELISA cut-offs corresponding to a specificity of 95%. Our results demonstrated that a combination of antigen-detection (CPS-LFI) and antibody-detection (Hcp1-ELISA or OPS-ELISA) tests increased the sensitivity for diagnosis of melioidosis (68% or 63%, respectively) over any single test, while maintaining high specificity (95%). In case patients, positivity of the CPS-LFI was associated with a short duration of symptoms, severe infections (as measured by an organ failure assessment score), bacteraemia and mortality outcome, while positivity of Hcp1-ELISA was associated with a long duration of symptoms, non-bacteraemia and survival outcome. Based on our findings, we propose that point-of-care melioidosis diagnostic tests using combinations of antigen- and antibody-detection should be further developed and evaluated.

## Introduction

Melioidosis, an infectious disease caused by the Gram-negative bacterium *Burkholderia pseudomallei*, is endemic to and has high mortality in tropical developing countries [1]. The disease is estimated to affect 165,000 people and account for 89,000 deaths per year worldwide [2]. Naturally acquired infections result from exposure through skin inoculation, inhalation or ingestion of *B. pseudomallei*, which is commonly present in soils and surface water in tropical countries [1,2]. The disease is difficult to diagnose and treat. Patients commonly present with sepsis, which is a syndrome defined by a dysregulated host response to infection resulting in

significant organ dysfunction and death. Sepsis can be caused by any variety of agents, including bacteria, fungi, and viruses [3]. A recent study in Thailand showed that there were 7,126 culture-confirmed melioidosis patients diagnosed from 2012 to 2015 in 70 hospitals countrywide, and that 39% of them died [4].

Culture remains the mainstay and gold standard for melioidosis diagnosis. Culture positivity for *B. pseudomallei* from any clinical sample is a definitive diagnosis for melioidosis since the organism is never part of the normal human flora (i.e. 100% specificity). Unfortunately, culture takes from 2 to 7 days and has a sensitivity of only about 60% based on a model estimate [5], and requires both experienced microbiologists and strict laboratory safety procedures [6]. Serological tests using crude antigen preparations, such as the indirect hemagglutination assay (IHA), are neither sensitive nor specific, and have no role in the diagnosis of melioidosis in melioidosis-endemic regions [7]. Specificity of the IHA ranges from 68% to 72% in Thailand using a cut-off of 1:160 according to the Thai standard [8,9] and from 75% to 91% in Australia and Papua New Guinea using a cutoff of 1:40 according to the Australian standard [10,11].

Misuse of diagnostic tests with low specificity, such as the IHA can lead to misdiagnoses, and thus, a lack of public health responses against melioidosis in Thailand [12]. Melioidosis has been a notifiable disease in Thailand since 2001, and a positive IHA result has been one of the criteria used to diagnose and report melioidosis cases in the country. This has led to a high number of IHA false-positive melioidosis cases with close to zero mortality being reported to the national notifiable disease surveillance system [12]. The incorrect mortality (ranging from 0.1 to 0.5%) has led to a false sense of security among people and policy makers, and limited prioritization by Ministry of Public Health [12]. In addition, misdiagnosis of melioidosis by using IHA results alone may lead to overuse of antibiotics that are effective against *B. pseudomallei* and place patients at risk of avoidable adverse drug reactions.

An increasing number of non-culture-based diagnostic tests for melioidosis are being developed and evaluated. Numerous PCR assays have been developed, but none are routinely used for clinical diagnosis in endemic areas because they are not cost-effective and their sensitivity is limited [1,7]. Recently, a lateral flow immunoassay (LFI) that detects the *B. pseudomallei* capsular polysaccharide (CPS) has been developed [13]. The test seems to have high specificity but limited sensitivity, particularly for blood samples [14–18]. It is possible that using a CPS-LFI in both serum and all non-blood specimens collected systematically from melioidosis suspected patients could increase the sensitivity of the combination further. Additionally, enzyme-linked immunosorbent assays (ELISA) that detect specific IgG antibodies against hemolysin co-regulated protein (Hcp1) or Type A O-polysaccharide (OPS) have been developed. An evaluation in northeast Thailand found that both Hcp1 and OPS-ELISA exhibited sensitivities and specificities ranging from 72% to 83% and 95% to 100%, respectively [19]. Assessment of a rapid immunochromatography test (ICT) that detects specific antibodies against Hcp1 has also reported a sensitivity of 88.3% in melioidosis patients in northeast Thailand and a specificity of 86.1% in Thai healthy donors [20].

We previously demonstrated that the CPS-LFI had limited sensitivity (31.3%) but high specificity (98.8%) for diagnosing melioidosis among patients presenting with community-acquired infections or sepsis in northeast Thailand when stored serum were tested [15]. Here, we evaluated combinations of the CPS-LFI and Hcp1- or OPS-ELISA for diagnosis of melioidosis. We hypothesized that a combination of antigen-antibody diagnostic tests may increase the sensitivity of melioidosis diagnosis over individual tests while preserving high specificity. The CPS-LFI was selected as an antigen-detecting diagnostic test under evaluation because it is in a point-of-care test that can be used to diagnose melioidosis in 15 minutes, has high specificity, and appears promising for resource-limited melioidosis-endemic settings [14–18]. Hcp1

and OPS-ELISA was selected as an antibody-detecting diagnostic test under evaluation because of its promising sensitivity and specificity [19].

## Material and methods

### Ethics statement

This study was conducted in full compliance with the principles of good clinical practice (GCP), and the ethical principles of the Declaration of Helsinki. The study protocol and related documents were approved by Sunpasitthiprasong Hospital Ethics Committee (039/2556), the Ethics Committee of the Faculty of Tropical Medicine, Mahidol University (MUTM2012-024-01), the University of Washington Institutional Review Board (42988) and the Oxford Tropical Research Ethics Committee at the University of Oxford (OXTREC172-12). Written, informed consent was obtained from participants prior to enrollment.

### Study population

We conducted a prospective observational (non-interventional) study of community-acquired infection and sepsis in Sunpasitthiprasong Hospital, Ubon Ratchathani province, northeast Thailand. From March 2013 to February 2017, we enrolled 5,001 adult patients (≥18 yr of age) who were admitted with a primary diagnosis of suspected or documented infections as determined by the attending physician, were within 24 h of hospital admission and had at least three sepsis diagnostic criteria documented in their medical record [21]. We excluded patients who were suspected of having hospital-acquired infections determined by the attending physician, had a hospital stay within 30 days prior to this admission or were transferred from another hospital with a total duration of hospitalization >72 hours. Organ dysfunction was determined by a modified Sequential (sepsis-based) Organ Failure Assessment (SOFA) score on admission as previously described [21]. 28-day mortality data were collected via telephone contact if subjects were no longer hospitalized and had been discharged alive [21]. Blood was drawn from all patients at the time of enrolment for culture and polymerase chain reaction (PCR) and serum samples were frozen at -80°C.

Patients who were culture positive for *B. pseudomallei* from any clinical specimens were selected as cases. Patients with blood cultures positive for *Staphylococcus aureus*, *Escherichia coli* or *Klebsiella pneumoniae*, or those patients testing positive for malaria or dengue via PCR assays, were selected as controls. Dengue and malaria were diagnosed by a nested PCR assay as described previously [22,23].

### CPS-LFI

The Active Melioidosis Detect LFI used in this study was developed by InBios (Seattle, WA, USA; lot no. WJ1222) as a research use only device [13]. Results from a study using this CPS-LFI alone were previously published [15], and used in this study. The result of this test was dichotomous; all weakly positive results were considered as positive because most weakly positive results were from culture-confirmed melioidosis cases [15].

### Hcp1-ELISA and OPS-ELISA

ELISA using Hcp1 or OPS antigens were performed essentially as previously described [19, 24]. Briefly, *B. pseudomallei* LPS Type A was extracted from the select agent excluded strain RR2808 (capsule mutant) using a modified hot aqueous-phenol method and purified O-polysaccharide (OPS) was then obtained via acid hydrolysis and gel permeation chromatography as previously described [25]. Hcp1 was obtained using recombinant DNA techniques and

purified as previously described [19]. For ELISA procedures, the microtiter plates were prepared by using optimized antigen concentrations of 2.5 μg/mL for Hcp1 and 1 μg/mL for OPS (50 μl per well for coating) and plates were incubated at 4°C for overnight. The plates were then washed 4 times with wash buffer using a Hydrospeed microplate washer followed by blocking with 5% skim milk in PBS at 37°C for 2 hours and were further washed as described above. Fifty microliters of serum diluted (1:2000) was added to wells and incubated at room temperature for 30 min. After washing, 50 μl of 1:2000 horseradish peroxidase-conjugated rabbit antihuman IgG (Dako, Denmark) was added and incubated at room temperature for 30 min. After last washing, 50 μl of TMB substrate solution (Invitrogen, USA) was added and incubated at room temperature for 15 min. The reaction was then stopped with 50 μl of 1N HCl and followed by the optical density (OD) measurement. All ELISAs were conducted in duplicate, and the absorbance values (optical density [OD]) were determined at a wavelength of 450 nm using a microtiter plate reader [19]. Pooled melioidosis sera (5 patients with culture-confirmed melioidosis) and pooled healthy sera (5 healthy Thai donors) were used as positive and negative sera controls for ELISAs. All ELISAs were conducted on specimens frozen at the time of patient enrollment [21], not on freeze-thaw specimens from any previous studies.

## Statistical analysis

Data were summarized with medians and interquartile ranges (IQR) for continuous measures, and proportions for discrete measures. IQRs are presented in terms of 25th and 75th percentiles. Continuous variables and proportions were compared between groups using Kruskal Wallis tests and Chi-square tests, respectively. All data in box plots are presented as $25^{th}$ and $75^{th}$ percentile boundaries in the box with the median line within the box; the whiskers indicate the $10^{th}$ and $90^{th}$ percentiles. A receiver operating characteristic (ROC) curve was created to monitor the shifting of the positive cut-off value of true-positive (sensitivity) and false positive (1-specificity) rates. Areas under the ROC curves (AUROCC) were compared using a non-parametric method. The lowest OD cut-off values that gave a specificity of 95% were selected. The sensitivity of diagnostic tests was defined as the proportion of melioidosis case patients who had positive test results. The specificity of diagnostic tests was defined as the proportion of control patients who had negative test results. The McNemar exact test was used to compare the sensitivity and specificity between tests.

We also explored the performance of these tests among case and control patients with different durations of symptoms, modified SOFA score, blood culture results and mortality outcome. The p values for trends were generated using the non-parametric test for trend across ordered groups. The Spearman correlation coefficient was used to explore the association between the OD value of ELISA and continuous variables. All analyses were performed using Stata version 14 (Stata Corp LP, College Station, TX, USA) and Prism 8 Statistics (GraphPad Software Inc, La Jolla, CA).

## Results

### Study participants

From March 2013 to February 2017, 5,001 adult patients presenting with community-acquired infections or sepsis were enrolled and followed for 28 days. A total of 193 patients were culture positive for *B. pseudomallei* and thus included as cases. Another 544 patients who were blood culture positive for *E. coli* (n = 189), *K. pneumoniae* (n = 27) and *S. aureus* (n = 53), or PCR positive for malaria (n = 152) and dengue (n = 123) were included in this study as controls. Serum was not available for one culture-confirmed melioidosis case. Serum of 42 patients with other confirmed diagnoses were not previously tested for CPS-LFI and were excluded from

**Table 1. Sensitivity and specificity of the CPS-LFI, Hcp1-ELISA, OPS-ELISA and the combinations of the CPS-LFI and Hcp1-ELISA or OPS-ELISA.**

| Assay | OD cut-off (of ELISA) | % Sensitivity (95% CI, N = 192) | % Specificity (95% CI, N = 502) |
|---|---|---|---|
| CPS-LFI | - | 31.1 (24.8–38.3) | 98.6 (97.1–99.4) |
| Hcp1-ELISA | 2.758 | 53.6 (46.3–60.9) | 95.0 (92.7–96.8) |
| OPS-ELISA | 2.839 | 48.4 (41.2–55.7) | 95.0 (92.7–96.8) |
| A combination of CPS-LFI and Hcp1-ELISA | 2.912 | 67.7 (60.6–74.3) | 95.0 (92.7–96.8) |
| A combination of CPS-LFI and OPS-ELISA | 3.100 | 63.0 (55.8–69.9) | 95.0 (92.7–96.8) |

the study. Therefore, a total of 192 culture-confirmed melioidosis cases and 502 controls were included in the analyses.

## Accuracy of the CPS-LFI, Hcp1-ELISA and OPS-ELISA

As previously reported [15], the CPS-LFI had a sensitivity of 31.3% (60/192 case patients were positive [95%CI 24.8 to 38.3]) and a specificity of 98.6% (495/502 control patients were negative [95%CI 97.1–99.4]) (Table 1). The median OD value of Hcp1-ELISA for the case patients was higher compared to control patients (median OD 2.922 [IQR 0.864–3.454]) vs. 0.299 [IQR 0.116–0.831], p<0.001) (Fig 1). The median OD value of OPS-ELISA for the case patients was also higher compared to control patients (median OD 2.695 [IQR 0.847–3.451]) vs. 0.604 [IQR 0.251–1.348], p<0.001). The AUROCCs of Hcp1-ELISA and OPS-ELISA were not significantly different (0.80 vs. 0.78, p = 0.12) (Fig 2).

Using a positive cut-off OD of 2.758 for Hcp1-ELISA to achieve a specificity of 95.0% (477/502 control patients were negative), Hcp1-ELISA had a sensitivity of 53.6% (103/192 case patients were positive; Tables 1 and S1). Using a positive cut-off OD of 2.839 for OPS-ELISA to achieve a specificity of 95.0% (477/502 control patients were negative), the OPS-ELISA had a sensitivity of 48.4% (93/192 case patients were positive).

## Accuracy of a combination of the CPS-LFI and Hcp1-ELISA or OPS-ELISA

To achieve a specificity equal to or higher 95% for the combination of antigen-detection and antibody-detection diagnostic tests, we found that the positive cut-off OD selected for the ELISAs needed to be higher than using the ELISAs alone (Tables 1, S1 and S2). Using a positive cut-off OD of 2.912 for Hcp1-ELISA to achieve a specificity of 95.0% (477/502 control patients were negative), the combination of the CPS-LFI and Hcp1-ELISA had a sensitivity of 67.7% (130/192 case patients were positive). Of 130 case patients with a positive test result, 33 (25.4%) were positive with the CPS-LFI alone, 70 (53.9%) were positive with Hcp1-ELISA alone, and 27 (20.8%) were positive with both tests.

Sensitivity of the CPS-LFI and Hcp1-ELISA combination was higher than that of CPS-LFI alone (67.7% vs. 31.3%, p<0.001) and that of Hcp1-ELISA alone (67.7% vs. 53.6%, p<0.001). A similar phenomenon was also observed for the CPS-LFI and OPS-ELISA combination. Sensitivity of the CPS-LFI and Hcp1-ELISA combination was not significantly different than that of CPS-LFI and OPS-ELISA combination (67.7% vs. 63.0%, p = 0.16).

## Sensitivity of diagnostic tests in melioidosis case patients by the duration of symptoms

Of 192 case patients, 44, 64, 44 and 40 reported having symptoms prior to admission for 1–2 days, 3–6 days, 7–13 days and ≥14 days, respectively. There was a trend showing that the sensitivity of the CPS-LFI decreased from 36.4% (16/44 case patients were positive) in patients with duration of symptoms for 1–2 days to 17.5% (7/40 case patients were positive, p = 0.004)

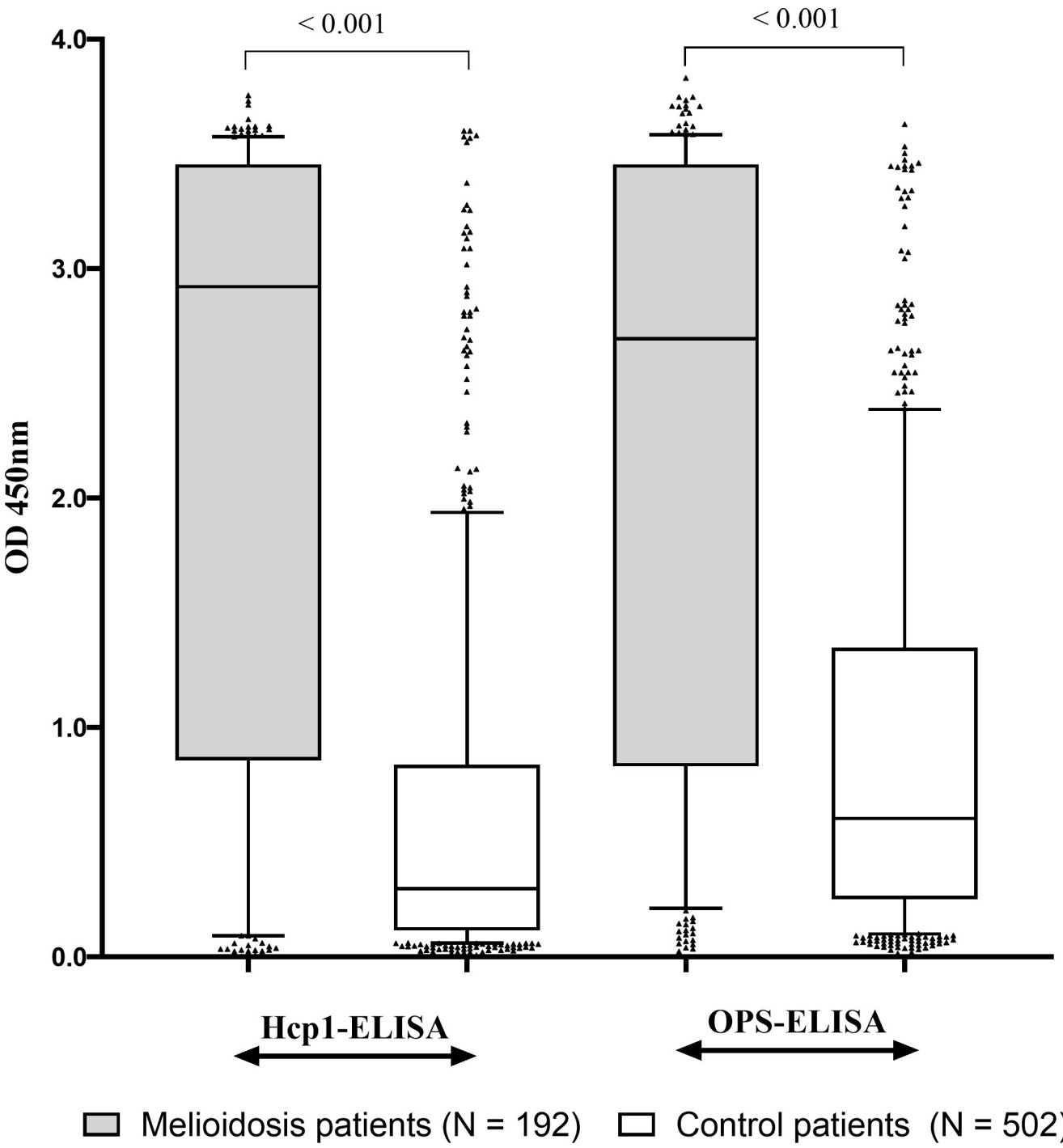

**Fig 1. Results of Hcp1-ELISA and OPS-ELISA from culture-confirmed melioidosis cases and controls**[*]. [*] Controls were patients whose blood culture were positive for *Staphylococcus aureus*, *Escherichia coli* or *Klebsiella pneumoniae* or those who were malaria or dengue polymerase chain reaction assay positive. Box plots represent $25^{th}$ and $75^{th}$ percentile boundaries in the box with the median line within box; the whiskers indicate the $10^{th}$ and $90^{th}$ percentiles. The plots show optimal density (OD) 450 of each antigen from serum samples collected within 24 hours of hospital admission from culture-confirmed melioidosis cases and controls.

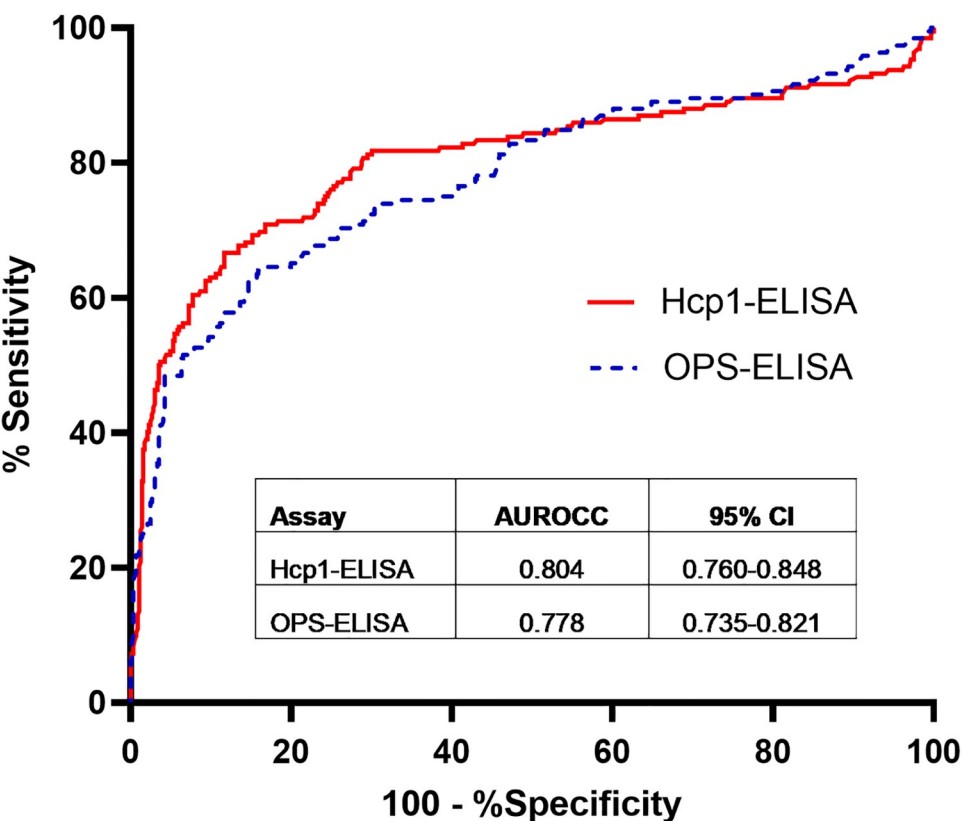

**Fig 2. Receiver operating characteristic curves (ROC) plots of Hcp1-ELISA and OPS-ELISA**\*. \*Areas under the ROC curves (AUROCC) for Hcp1-ELISA and OPS-ELISA were calculated from the optimal density (OD) from serum samples collected within 24 hours of hospital admission from culture-confirmed melioidosis cases and controls.

in patients with symptoms for ≥14 days (Fig 3 and S3 Table). The sensitivity of Hcp1-ELISA correspondingly increased from 45.5% (20/44 case patients with symptoms for 1–2 days were positive) to 77.5% (31/40 case patients with symptoms for ≥14 days were positive, p<0.001). We did not observe a trend in the sensitivity of the CPS-LFI and Hcp1-ELISA combination by the duration of symptoms (p = 0.15); the sensitivity ranged from 65.9% (29/44 cases patients were positive) in patients with duration of symptoms for 1–2 days to 80.0% (32/40 cases patients were positive) in patients with symptoms for ≥14 days. A similar pattern was also observed for the OPS-ELISA (Fig 4).

## Sensitivity of diagnostic tests in melioidosis case patients by modified SOFA score

Of 192 case patients, 42, 22, 41 and 87 had a modified SOFA score of 0–1, 2–3, 4–5 and ≥6, respectively. There was a trend showing that the sensitivity of the CPS-LFI increased from 7% (3/42 case patients were positive) in patients with a modified SOFA score of 0–1 to 49% (43/87 case patients were positive) in patients with a modified SOFA score of ≥6 (p<0.001) (S3 Table). The sensitivity of Hcp1-ELISA alone correspondingly decreased from 66.7% (28/42 case patients with modified SOFA score of 0–1 were positive) to 47.1% (41/87 case patients with modified SOFA score ≥6 were positive, p = 0.03). We did not observe a trend in the sensitivity of the CPS-LFI and Hcp1-ELISA combination by modified SOFA score (p = 0.62). A similar pattern was also observed for the OPS-ELISA.

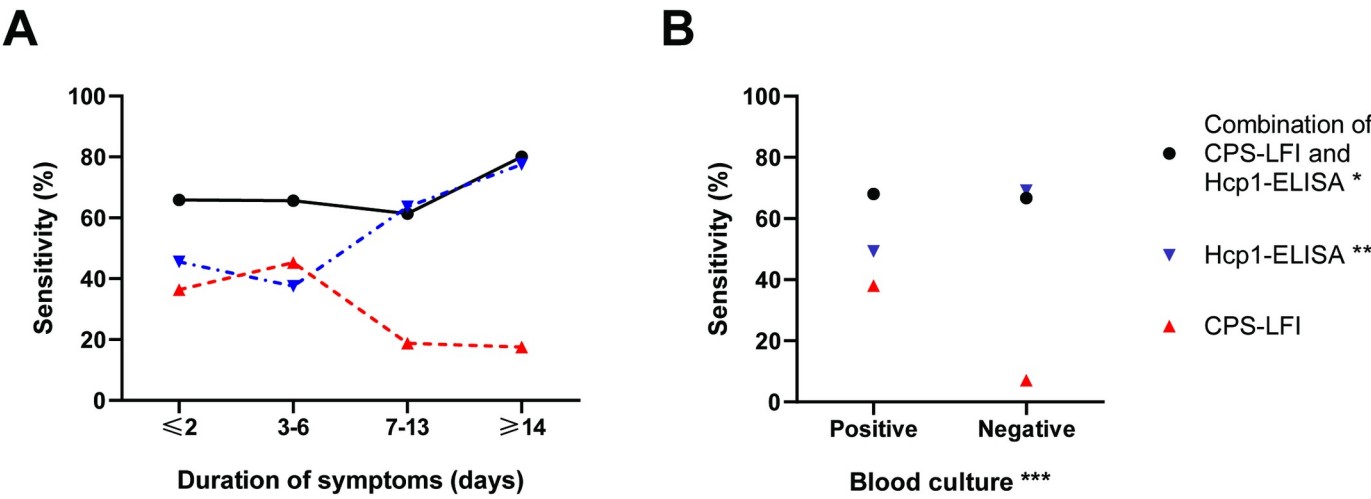

**Fig 3. Sensitivity of the combination of the CPS-LFI and Hcp1-ELISA, Hcp1-ELISA alone and the CPS-LFI alone.** (A) by duration of symptoms prior to admission and (B) by blood culture results in 192 melioidosis cases. * using OD cut-off value at a specificity of 95% (OD 2.912) ** using OD cut-off value at a specificity of 95% (OD 2.758) *** Blood culture positive for *B. pseudomallei*.

### Sensitivity of diagnostic tests in melioidosis case patients by bacteraemia status

Of 192 case patients, 150 (78.1%) were blood culture positive for *B. pseudomallei*. As previously reported [15], the sensitivity of the CPS-LFI was higher in patients with positive blood culture (38.0% [57/150]) than in those with negative blood culture (7.1% [3/42], p<0.001) (Fig 3). The sensitivity of Hcp1-ELISA alone was lower in patients with positive blood culture (49.3% [74/150]) than in those with negative blood culture (69.1% [29/42], p = 0.02). Overall, the sensitivity of the CPS-LFI and Hcp1-ELISA combination was not associated with blood culture results (p = 0.87), ranging from 68.0% [102/150] in patients with positive blood culture to 66.7% [28/42] in patients with negative blood cultures. A similar pattern was also observed for OPS-ELISA (Fig 4).

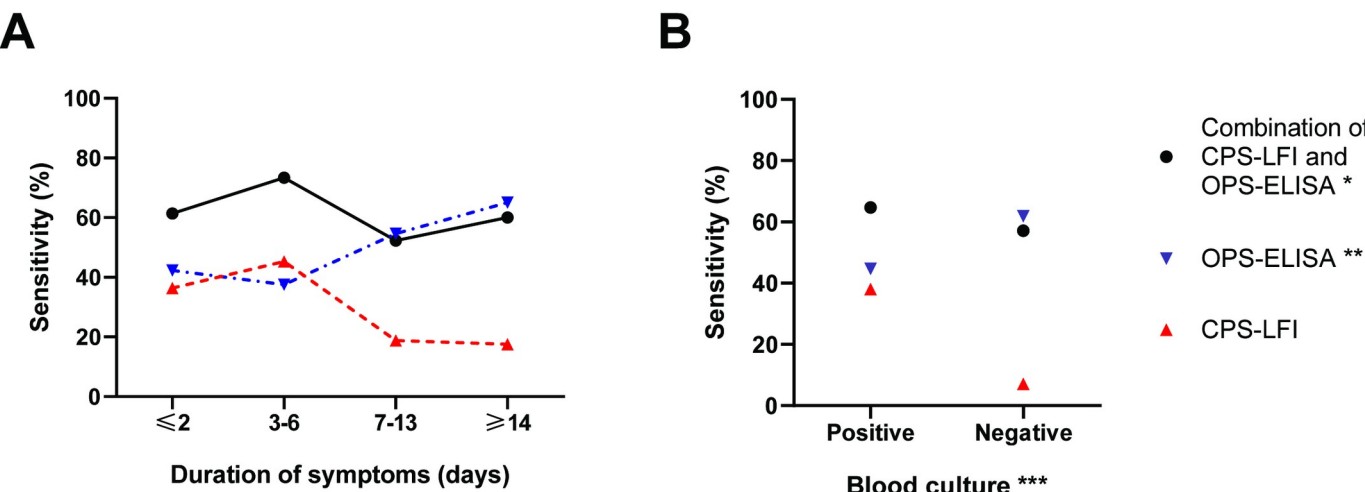

**Fig 4. Sensitivity of the combination of the CPS-LFI and OPS-ELISA, OPS-ELISA alone and the CPS-LFI alone.** (A) by duration of symptoms prior to admission and (B) by blood culture results in 192 melioidosis cases. * using OD cut-off value at a specificity of 95% (OD 3.100) ** using OD cut-off value at a specificity of 95% (OD 2.839) *** Blood culture positive for *B. pseudomallei*.

### Association between diagnostic test results and 28-days mortality in melioidosis case patients

Of 192 melioidosis case patients, 99 (51.6%) died within 28 days of hospital admission. Patients with positive CPS-LFI were more likely to die than those with negative CPS-LFI results (72% [43/60] vs. 42% [56/132], p<0.001) (S3 Table). Patients with positive Hcp1-ELISA results were associated with lower mortality than those with negative Hcp1-ELISA results (43.7% [45/103] vs. 60.7% [54/89], p = 0.02). More specifically, the median OD value for Hcp1-ELISA in the 99 case patients who died was significantly lower than that of the 93 case patients who survived up to 28 days after hospital admission (2.448 [IQR 0.729–3.300] vs. 3.273 [IQR 1.342–3.494], p = 0.009).

Positive OPS-ELISA results were not significantly associated with lower mortality than those with negative OPS-ELISA results (47.3% [44/93] vs. 55.6% [55/99], p = 0.25). The median OD value for OPS-ELISA in 99 cases patients who died was not significantly lower than that of 93 case patients who survived up to 28 days after hospital admission (2.342 [IQR 0.801–3.331] vs. 2.966 [IQR 1.052–3.488], p = 0.18).

### Specificity of diagnostic tests in different groups of control patients

Specificity of the CPS-LFI was not significantly different among groups of control patients (p = 0.86, S4 Table). However, specificity of Hcp1-ELISA was highest among patients with PCR positive results for dengue (99.2%; 122/123) and lowest among patients with positive blood cultures for *S. aureus* (90.0%; 18/20) or PCR positive results for malaria (90.1%; 136/151, p = 0.003). This pattern was also observed with OPS-ELISA (p = 0.04). Specifically, the median OD value of Hcp1-ELISA and OPS-ELISA were highest among patients who were PCR positive for malaria (p = 0.001 and p = 0.002, respectively, S5 Table), while the low specificity among patients who were blood culture positive for *S. aureus* were likely caused by two outliers with high OD values. Additionally, there was some evidence that the specificity of Hcp1-ELISA and OPS-ELISA decreased in control patients with higher modified SOFA score (p = 0.08 and p = 0.02, respectively, S4 Table). A very weak positive correlation between modified SOFA scores and the OD values of Hcp1-ELISA (rho 0.14, p = 0.002) or OPS-ELISA (rho 0.15, p<0.001) was also observed.

Of 502 control patients, 51 (10.2%) died within 28 days of hospital admission. An association between diagnostic test results and 28-day mortality of control patients was not observed (S4 Table).

### Discussion

This study of patients hospitalized within 24 hours with community-acquired infection and sepsis at a referral hospital in northeast of Thailand demonstrates that a combination of the CPS-LFI and Hcp1-ELISA or CPS-LFI and OPS-ELISA increased the sensitivity of melioidosis diagnosis over any of the three tests alone while maintaining high specificity. This is an important advancement over current insensitive or protracted diagnostic strategies for disease with a nearly 40% case fatality rate in northeast of Thailand.

Although the CPS-LFI is in a format that can be readily used as a rapid diagnostic test, it is still in the development and evaluation phase and not yet commercially available. A rapid immunochromatography test (ICT) using Hcp1 as the target antigen has been developed and has a strong agreement with Hcp1-ELISA results [20]. The Hcp1-ICT is currently available for research use only from Mahidol University. The increased diagnostic accuracy of the

combination of antigen and antibody detection tests indicates that these tests will be valuable in optimizing the care of melioidosis patients.

Additional findings are that in case patients, positivity of CPS-LFI is associated with patients with shorter duration of symptoms, higher modified SOFA score, bacteraemia and 28-day mortality outcome, while positivity of Hcp1-ELISA is associated with longer duration of symptoms, lower modified SOFA score, non-bacteraemia and 28-day survival outcome. These features of the tests may further support physicians in making triage decisions, resource allocation and antibiotic stewardship in resource-limited melioidosis-endemic regions.

The benefits of combining antigen and antibody-detecting diagnostic tests for melioidosis are consistent with combination diagnostic tests that are recommended for some infectious diseases such as dengue infection. Currently, point-of-care tests for dengue diagnosis include NS1 and IgM-based tests [26]. NS1 antigen can be detected in dengue patients within the first few days after the onset of illness, while IgM is still not detectable. By day 5 after the onset of illness, however IgM is detectable in 80% of dengue patients while NS1 may only be detectable in some patients [27]. This phenomenon is also observed in our study, in which the sensitivity of the antigen-detecting (CPS-LFI) test also declines in patients with longer duration of symptoms, while the sensitivity of the antibody-detecting (Hcp1- and OPS-ELISA) test increases.

Our defined cut-off values for Hcp1-ELISA and OPS-ELISA were higher than those used in the previous study [19]. This may be due to differences in study populations as well as timing of the specimen collections. Patient samples used in this study were obtained from a cohort of patients presenting with community-acquired infection and sepsis in northeast Thailand (Ubon-sepsis) whose blood was sampled within 24 hours of study hospital admission, while patient samples used in the previous study were obtained from multiple populations and different time points during illness (e.g. samples were obtained from case patients at a median of 5 days after hospital admission) [19]. Control patients in the previous study included Thai healthy donors, U.S. healthy donors, tuberculosis patients, scrub typhus patients and leptospirosis patients [19], and the OD values from those control patients were lower than what we observed from patients in this study who were blood culture positive for *E. coli*, *K. pneumoniae* or *S. aureus*, or PCR positive for malaria and dengue. Control patients enrolled in our study died within 28 days of hospital admission at a rate of 10%. Collecting blood specimens within 24 hours of admission can reduce the possibility of survival bias [28,29], in which only patients who survive up to a specific time point can be evaluated and enrolled into a study. For example, only patients who survive up to when results of bacterial culture are reported can be evaluated and enrolled into a study. The sensitivity of Hcp1-ELISA in this study (53.6%) was lower than that previously reported (83.0%) [19], which was due to the higher cut-off OD value required to give a specificity of 95.0% in the cohort setting.

The sensitivity of the CPS-LFI in this study using stored sera (31.3%) was higher than that recently reported from a study in India using whole blood (25.0% [2/8]) [17] and a study in Laos using stored sera (13.9% [5/36]) [18]. These differences could be caused by the use of different study populations, timing of specimen collection and generation of CPS-LFI test. It is also possible that melioidosis patients presenting in northeast Thailand have higher levels of CPS in their blood on admission and enrollment. If this is the case, these high CPS levels may be associated with higher mortality outcomes of melioidosis in northeast Thailand compared to those observed in India and Laos [17,18]. A previous study also found that the CPS-LFI appeared to perform better with blood that have been collected a few days prior to the collection of blood cultures that subsequently yield positive culture [14]. Further studies on factors associated with sensitivity and specificity of diagnostic tests for melioidosis are required.

This study has four strengths. First, cases and controls were drawn from a large prospective observational study of patients presenting with community-acquired infection and sepsis in

northeast Thailand which represented a real-world setting. Second, all serum samples were drawn within 24 hours of admission to the study hospital which is ideal for evaluating point-of-care diagnostic tests. This approach would also avoid a survival bias [28,29]. Third, the prospective study collected blood and other relevant clinical specimens for bacterial culture from every patient enrolled systematically. This allows us to evaluate the accuracy of new diagnostic tests based on culture positivity for *B. pseudomallei* from both blood and non-blood specimens with low sample selection bias [21]. Fourth, the study evaluated modified SOFA score on admission and followed all patients for 28-day mortality outcome.

A limitation of this study is that positive predictive and negative predictive values could not be estimated because of the case-control study design. While preserving high specificity of the combination, the sensitivity of the combination was only 67.7%. This means that this combination may still miss a moderate proportion of melioidosis patients. The CPS-LFI can detect the *B. pseudomallei* CPS in a wide range of clinical specimens, including sera, sputum, urine, pus and sterile fluid [17,18]. However, we could not evaluate the accuracy of the CPS-LFI in non-blood specimens as they were not systematically stored during the prospective study. It is possible that using a CPS-LFI in both serum and all non-blood specimens collected systematically from melioidosis suspected patients could increase the sensitivity of the combination further. Our study did not evaluate LPS type of *B. pseudomallei* isolated from the case patients. *B. pseudomallei* LPS can be categorized into typical type A, atypical types B1 and B2, and rough type to represent the difference of OPS. The OPS-ELISA used in the previous and this study was prepared from *B. pseudomallei* LPS type A [19,25], and may have limited sensitivity among patients infected with *B. pseudomallei* with atypical or rough LPS type. A high proportion of patients enrolled into the Ubon-sepsis cohort (71%) were transferred from district hospitals, smaller hospitals in the province and hospitals in other provinces. The performance of diagnostic tests could vary in different settings, and, therefore, generalizability of these findings may be limited. The sera were tested for CPS-LFI in 2017 [15] and tested for ELISAs in 2020. We acknowledge that different durations of specimen storage could affect the performance of diagnostic tests. We also acknowledge that ELISAs under evaluation are not point-of-care tests as we need to define optimal cutoffs for ELISA when used in combination.

In conclusion, our results indicate that the combination of antigen and antibody-detection diagnostic tests significantly improved the sensitivity of melioidosis diagnosis, compared to when the tests were used individually, while also maintaining high specificity. Retaining high specificity while achieving good sensitivity of diagnostic tests in the real-world setting is essential. We propose that rapid diagnostic tests for melioidosis based on the combination of antigen-antibody detection should be further studied and developed as implementation of these tests is likely to benefit tropical developing countries where melioidosis is endemic.

## Supporting information

**S1 Table. Sensitivity and specificity of Hcp1-ELISA and OPS-ELISA using different OD cut-off values.**
(DOCX)

**S2 Table. Sensitivity and specificity of a combination of the CPS-LFI and Hcp1-ELISA and a combination of the CPS-LFI and OPS-ELISA using different OD cut-off values.**
(DOCX)

**S3 Table. Diagnostic test results in different groups of melioidosis case patients.**
(DOCX)

**S4 Table. Diagnostic test results in different groups of control patients.** (DOCX)

**S5 Table. OD values of ELISA in different groups of control patients.** (DOCX)

## Acknowledgments

We are grateful to the patients and staff of Sunpasitthiprasong Hospital, and the Wellcome Trust-Oxford University-Mahidol University Tropical Medicine Research Program. We thank Praweennuch Watanachaiprasert, Kantiya Jirapornuwat, Mayura Malasit, Passaraporn Kesaphun, Chayamon Krainoonsing, Areeya Faosap, Yaowaret Dokket, Sukhumal Pewlaorng, Sayan Langla, Sineenart Sengyee and Taniya Kaewarpai for their clinical, laboratory and administrative support.

## Author Contributions

**Conceptualization:** Premjit Amornchai, Viriya Hantrakun, Vanaporn Wuthiekanun, Surasakdi Wongratanacheewin, Direk Limmathurotsakul.

**Data curation:** Viriya Hantrakun, Direk Limmathurotsakul.

**Formal analysis:** Premjit Amornchai, Viriya Hantrakun, Direk Limmathurotsakul.

**Funding acquisition:** Surasakdi Wongratanacheewin, T. Eoin West, Direk Limmathurotsakul.

**Investigation:** Premjit Amornchai, Gumphol Wongsuvan, Surasakdi Wongratanacheewin.

**Methodology:** Premjit Amornchai, Vanaporn Wuthiekanun, T. Eoin West, David P. AuCoin, Paul J. Brett, Mary N. Burtnick, Narisara Chantratitra, Direk Limmathurotsakul.

**Resources:** Surasakdi Wongratanacheewin, Prapit Teparrakkul, T. Eoin West, David P. AuCoin, Nicholas P. J. Day, Paul J. Brett, Mary N. Burtnick, Narisara Chantratitra.

**Supervision:** Viriya Hantrakun, Vanaporn Wuthiekanun, Narisara Chantratitra, Direk Limmathurotsakul.

**Validation:** Premjit Amornchai, Gumphol Wongsuvan, Surasakdi Wongratanacheewin, Direk Limmathurotsakul.

**Writing – original draft:** Premjit Amornchai, Vanaporn Wuthiekanun, Direk Limmathurotsakul.

**Writing – review & editing:** Premjit Amornchai, Viriya Hantrakun, Gumphol Wongsuvan, Vanaporn Wuthiekanun, Surasakdi Wongratanacheewin, Prapit Teparrakkul, T. Eoin West, David P. AuCoin, Nicholas P. J. Day, Paul J. Brett, Mary N. Burtnick, Narisara Chantratitra, Direk Limmathurotsakul.

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
