## [Decision Letter · Decision Letter 0]

4 Jul 2021

Dear Amornchai,

Thank you very much for submitting your manuscript "Evaluation of antigen-detecting and antibody-detecting diagnostic test combinations for diagnosing melioidosis" for consideration at PLOS Neglected Tropical Diseases. As with all papers reviewed by the journal, your manuscript was reviewed by members of the editorial board and by several independent reviewers. In light of the reviews (below this email), we would like to invite the resubmission of a significantly-revised version that takes into account the reviewers' comments. 

We cannot make any decision about publication until we have seen the revised manuscript and your response to the reviewers' comments. Your revised manuscript is also likely to be sent to reviewers for further evaluation.

Sincerely,

Husain Poonawala

Associate Editor

Anna Ralph

Deputy Editor

Reviewer's Responses to Questions

**Key Review Criteria Required for Acceptance?**

**Methods**

-Are the objectives of the study clearly articulated with a clear testable hypothesis stated?

-Is the study design appropriate to address the stated objectives?

-Is the population clearly described and appropriate for the hypothesis being tested?

-Is the sample size sufficient to ensure adequate power to address the hypothesis being tested?

-Were correct statistical analysis used to support conclusions?

-Are there concerns about ethical or regulatory requirements being met?

Reviewer #1: No changes needed

Reviewer #2: Some experiments lack appropriate controls

Some of the methodology is written very briefly and requires the reader to look for the referenced protocol. Not all journals are easily accessible especially for those in countries where melioidosis is endemic.

The sample size is adequate but I question the comparison that was done on the same samples but 3 years (?) apart with no statistical analysis between tests.

Reviewer #3: (No Response)

**Results**

-Does the analysis presented match the analysis plan?

-Are the results clearly and completely presented?

-Are the figures (Tables, Images) of sufficient quality for clarity?

Reviewer #1: Yes

Reviewer #2: Yes but the authors should refer to comments within the attachment.

Reviewer #3: (No Response)

**Conclusions**

-Are the conclusions supported by the data presented?

-Are the limitations of analysis clearly described?

-Do the authors discuss how these data can be helpful to advance our understanding of the topic under study?

-Is public health relevance addressed?

Reviewer #1: Yes

Reviewer #2: The limitations are identified but taken together and including other limitations not specifically mentioned in this submission makes it somewhat difficult to appreciate the novelty and applicability of the data in a public health setting, particularly in melioidosis endemic countries.

Reviewer #3: (No Response)

**Editorial and Data Presentation Modifications?**

Reviewer #1: None needed

Reviewer #2: (No Response)

Reviewer #3: (No Response)

**Summary and General Comments**

Reviewer #1: This paper uses a subset of a cohort of patients from a sepsis study in NE Thailand to evaluate a combination of antibody and antigen detection in serum for the diagnosis of melioidosis and demonstrates the superior sensitivity of this combined approach compared with individual tests, although this was still only 67.7% even when compared with the imperfect gold standard of culture. Whilst I agree that this is an important development, the fact that it will still miss many patients with melioidosis shopuld be given greater prominence.

It is implicit that the tests were only conducted on serum, although the CPS antigen detection test can also be used on other samples, such as pus, sputum and urine, and in such samples it may have higher sensitivity, but this is not made explicit until the discussion, so this should be made clear earlier in the manuscript.

There are a few minor typos that need correcting:

1. Line 93: change 'accounts' to 'account'.

2. Line 123: change 'placing' to 'place'.

3. Line 131: change 'detects' to 'detect'.

4. Line 432: change 'features' to 'has'.

Reviewer #2: Please refer to the attachment

Reviewer #3: Dear Editor,

This manuscript describes the use of 3 immunological based assays to diagnose melioidosis, a fatal tropical disease, in patients in Northeast Thailand. Their findings have suggested that the assays would be more effective if they were used in combinations: CPS-LFI and Hcp1-ELISA, or CPS-LFI and OPS-ELISA. These combinations improved the sensitivity, while the specificity of the combinations remained at 95% or more. However, the adjusted OD cut-off values of ELISA need to be used to achieve the acceptable sensitivity and specificity when the assays were used in a combination. This is a good finding since these 3 assays have been validated in multiple studies earlier. However, the manuscript is not well written, and the calculations may be inaccurate. Here are some comments:

Major comments:

1. It was not mentioned how the sensitivity or the specificity was calculated. Based on their findings, it is unlikely that the authors used the false positive value to calculate the sensitivity, and the false negative value to calculate the specificity. Please note that the sensitivity of an assay is the proportion of the diseased people correctly identified, while the specificity of same assay is the proportion of non-diseased people correctly identified. 

% Sensitivity = {a / (a + b)} × 100

% Specificity = {d / (d + c)} × 100 

a, true positive (assay positive, culture positive)

b, false positive (assay positive, culture negative)

c, false negative (assay positive, culture negative)

d, true negative (assay negative, culture negative)

Did the authors observe the false positive or false negative results at all? If they did, these values need to be included in formulas above. The authors will need to mention in the manuscript if there were false positive or false negative results. 

2. It has been known that B. pseudomallei has diverse LPS types. Is OPS-ELISA able to detect immunoglobulin responses in sera from patients who were infected by B. pseudomallei strains with other LPS/O-antigen types? Has this been tested? Since the atypical LPS types (B, or rough) have been detected in B. pseudomallei strains from Thailand, targeting only the LPS type A by this assay may limit the efficacy of this assay. The authors will need to discuss this limitation.

3. Are both Hcp1-ELISA and OPS-ELISA detecting igG, IgM, or total IgG? It is worth mentioning it, otherwise the readers have to check with ref. 19 and 20 for more details. Since both assays have been compared in ref. 19, and the Hcp1-ELISA was the best one, why did the authors compare both assays again in this study? Were the patients' sera used in this study from the same cohort used in ref. 19? 

4) It was not clear if the authors tested all these 3 assays at the same time in this study, or only used the results from previous studies for statistical analysis? Please clarify this since the authors cited ref. 15 and 19 in most parts of their methods. 

Other comments:

1. In the last statement of the Author Summary, the authors should suggest whether the combination, CPS-LFI and Hcp1-ELISA, or CPS-LFI and OPS-ELISA, is further evaluated for the point-of-care diagnosis of melioidosis rather than saying in general that "using combinations of antigen- and antibody-detection should be further developed and evaluated". 

2.Line: 126-128: Did the authors mean the sensitivity of PCR was limited in detecting B. pseudomallei in clinical specimens?

PLOS authors have the option to publish the peer review history of their article (what does this mean?). If published, this will include your full peer review and any attached files.

Reviewer #1: Yes: Prof. David Dance

Reviewer #2: No

Reviewer #3: No
---

## [Decision Letter · Decision Letter 1]

27 Sep 2021

Dear Ms Amornchai,

We are pleased to inform you that your manuscript 'Evaluation of antigen-detecting  and antibody-detecting diagnostic test combinations for diagnosing melioidosis' has been provisionally accepted for publication in PLOS Neglected Tropical Diseases.

Best regards,

Husain Poonawala

Associate Editor

Paul J. Brindley

Co-Editor-in-Chief

Reviewer's Responses to Questions

**Key Review Criteria Required for Acceptance?**

**Methods**

-Are the objectives of the study clearly articulated with a clear testable hypothesis stated?

-Is the study design appropriate to address the stated objectives?

-Is the population clearly described and appropriate for the hypothesis being tested?

-Is the sample size sufficient to ensure adequate power to address the hypothesis being tested?

-Were correct statistical analysis used to support conclusions?

-Are there concerns about ethical or regulatory requirements being met?

Reviewer #1: (No Response)

Reviewer #2: Yes, the previous comments have been addressed.

Reviewer #3: (No Response)

**Results**

-Does the analysis presented match the analysis plan?

-Are the results clearly and completely presented?

-Are the figures (Tables, Images) of sufficient quality for clarity?

Reviewer #1: (No Response)

Reviewer #2: Yes

Reviewer #3: (No Response)

**Conclusions**

-Are the conclusions supported by the data presented?

-Are the limitations of analysis clearly described?

-Do the authors discuss how these data can be helpful to advance our understanding of the topic under study?

-Is public health relevance addressed?

Reviewer #1: (No Response)

Reviewer #2: The limitations can be re-emphasised with caveats on the samples used

Reviewer #3: (No Response)

**Editorial and Data Presentation Modifications?**

Reviewer #1: (No Response)

Reviewer #2: Grammatical errors that need to be corrected

The figshare link on line 228 is not available

Reviewer #3: (No Response)

**Summary and General Comments**

Reviewer #1: (No Response)

Reviewer #2: Most of my comments and suggestions from the previous review have been sufficiently addressed.

However, the improvement in sensitivity demonstrated by the combination tests is still subjective because the individual test data was obtained at much earlier dates. How the cut-offs for controls and samples were determined is still not clear.

I suggest that the authors have to emphasise the limitations of the study design and samples and provide caveats on the feasibility of the combination tests for routine clinical diagnosis, particularly in rural hospitals, as the interpretation in association with when the patient presents at the hospital can be somewhat subjective.

Reviewer #3: Dear Editor,

The revised manuscript has most improvements and is well discussed. The authors have addressed my comments and concerns.

PLOS authors have the option to publish the peer review history of their article (what does this mean?). If published, this will include your full peer review and any attached files.

Reviewer #1: No

Reviewer #2: No

Reviewer #3: **Yes: **Apichai Tuanyok

---

## [Editor Report · Acceptance letter]

20 Oct 2021

Dear Ms Amornchai,

We are delighted to inform you that your manuscript, "Evaluation of antigen-detecting  and antibody-detecting diagnostic test combinations for diagnosing melioidosis," has been formally accepted for publication in PLOS Neglected Tropical Diseases.

Best regards,

Shaden Kamhawi

co-Editor-in-Chief

Paul Brindley

co-Editor-in-Chief
